# Short-End Injection Capillary Electrophoresis and Multivariate Analysis for Simultaneous Determination of Heavy Metals in *Passiflora incarnata* Tea

**DOI:** 10.3390/ijerph192315994

**Published:** 2022-11-30

**Authors:** Aline Klassen, Rafael Ferreira Fernandes, Débora Cristina de Oliveira, Maria Patrícia do Nascimento, Marcella Matos Cordeiro Borges, Marcone Augusto Leal de Oliveira, Leandro Augusto Calixto, Keyller Bastos Borges

**Affiliations:** 1Instituto de Ciências Ambientais, Químicas e Farmacêuticas, Universidade Federal de São Paulo (UNIFESP), Campus Diadema, Rua São Nicolau, 210, Centro, Diadema 09913-030, SP, Brazil; 2Departamento de Química, Instituto de Ciências Exatas, Universidade Federal de Juiz de Fora (UFJF), Campus Universitário, Rua José Lourenço Kelmer, s/n, Martelos, Juiz de Fora 36036-900, MG, Brazil; 3Departamento de Ciências Naturais, Universidade Federal de São João del-Rei (UFSJ), Campus Dom Bosco, Praça Dom Helvécio 74, Fábricas, São João del-Rei 36301-160, MG, Brazil

**Keywords:** design of experiments, *Passiflora incarnata* tea, elemental impurities, short-end injection

## Abstract

An ultra-fast method for the simultaneous determination of heavy metals in *Passiflora incarnata* tea by capillary electrophoresis (CE) using a short-end injection combined with multivariate analysis was proposed. Separation was conducted by hydrodynamic injection (5 s at 0.5 psi) using the short-end injection procedure in a fused uncoated silica capillary (50 cm total length, 10.2 cm effective length, 50 µm i.d.) with separation time less than 2 min. An indirect UV detection at 214 nm was employed by using imidazole as a chromophore. The buffer used was 6 mmol/L hydroxybutyric acid (HIBA). The optimum conditions by full factorial with a central point were achieved by 18-crown-6 concentration (23.3 mmol L^−1^), voltage (+11.4 kV), methanol concentration (3.8%), and temperature (20 °C). The method showed good linearity (R^2^ > 0.998) for both Cd and Pb, inter-day precision of less than 14.49%, and an adequate limit of quantification only for Cd (LOQ < 0.5 µg mL^−1^ for Cd) based on the US Pharmacopeial Convention limit requirements for elemental impurities. After method validation, the method was applied to *Passiflora incarnata* tea samples from a local market. Furthermore, the developed method showed great potential for the determination of metals in other samples with proper sample preparation procedures.

## 1. Introduction

Humans have consumed medicinal plants for therapeutic purposes for a long time. Use tends to grow over the years in developed and developing countries [1]. Among medicinal plants, plants of the genus Passiflora stand out, which present secondary metabolites such as phenols, glycosylated flavonoids, and cyanogenic compounds. The plants of this genus are composed of more than 500 species and have been used as medicinal plants in Mexico, South America, Holland, and Italy, mainly for the treatment of anxiety control. *Passiflora incarnata* is considered the official species for herbal treatments since it has been mentioned in several pharmacopoeias [2]. Few human studies with this plant focus only on anxiety control, but studies indicate that their properties act on the inhibition of monoamine oxidase as well as activating γ-aminobutic acid receptors that are directly linked to neuropsychiatric disorders, some of which are anxiety and depression [2]. However, there are several different studies on treatment of opiate withdrawal, insomnia, attention deficit hyperactivity disorder, and even cancer. The medicinal properties of *Passiflora incarnata* related to the improvement of anxiety are mainly from the leaves and juice of the pulp of the fruit, where there is passiflorin, a natural sedative. The tea made from its leaves has a depurative, diuretic, and anti-inflammatory effect. Passion fruit is a source of carbohydrates, vitamins A, C, and B complex, as well as being rich in calcium, iron, and zinc minerals [2].

There is an imminent concern regarding the contamination of medicinal plants with heavy metals. In large centers, extractive collection occurs largely where the origin of the crop’s (soil, irrigation, and fertilizer) transport, storage, and hygiene are not known. Consequently, there are possible sources of contamination, mainly for the plants in need of water, minerals, and other nutrients [3,4]. Therefore, in the case of environmental contamination, heavy metals can accumulate in plant tissues. Humans are subject to contamination by heavy metals by ingesting both homemade and industrial plants or teas. There are reports of humans being contaminated with heavy metals after undertaking home treatments using Chinese and Indian medicinal plants [3,4]. Intoxication can be acute, chronic, or subchronic; it can also be neurotoxic, carcinogenic, mutagenic, or teratogenic; this depends on the amount ingested and on the type of metal impurity. Symptoms can present as biochemical disturbances in the body, gastrointestinal problems, tremor, convulsions, and paralysis, among other illnesses [5].

Some metals such as cadmium (Cd), lead (Pb), mercury (Hg), arsenic (As), and chromium (Cr), among others, can be carried to the environment by natural or anthropogenic routes in their organic and/or inorganic forms. Some heavy metals, such as iron, zinc, calcium, and magnesium, are considered of great importance for human health. These have an ideal amount for daily intake and their tolerance levels are much higher than toxic heavy metals such as Cd, Pb, among others [5]. Cd is very dangerous to humans since it is considered toxic in extremely low amounts. Depending on exposure, it can cause damage to the respiratory system once there is contamination by inhalation of contaminated dust or vapors, which can cause chest pain, failure of the lung tissue linings, emphysema, bronchiolitis, etc. In the kidneys, it can compromise renal function determined by tubular proteinuria. It can also cause increased blood pressure and myocardial dysfunction; in addition to all these problems, contamination with Cd can be associated with bone problems such as osteoporosis, osteomalacia, and even instant fracture [5]. Pb contamination comes from food, water, and inhalation. Its health problems in its ionic form cause problems with the gastrointestinal system and dysfunctions in the central nervous system and peripheral nervous system. It can also cause inhibition of hemoglobin synthesis, kidney dysfunction, joint pain, problems with the reproductive and cardiovascular system, and even blood in the urine [5].

The US Pharmacopeial Convention, an organization involved in helping protect patient safety and strengthen the global supply of quality medicines in the USA market, has established the chapter <2232> to aid the FDA to regulate elemental contaminants in dietary supplements. The analytical techniques required are either inductively coupled plasma–optical emission spectrometry (ICP-OES) or inductively coupled plasma mass spectrometry (ICP-MS) [6]. Moreover, some other analytical techniques, such as atomic absorption spectrometry (AAS), high performance liquid chromatography (HPLC), and neutron activation analysis (NAA) have been employed in the evaluation of herbal drugs, as well as in analysis of other matrices containing heavy metals [7].

However, some these techniques are limited due to them requiring sophisticated instruments and high operational costs. In recent years, capillary electrophoresis has been considered an interesting alternative technique to access inorganic ions due to its high versatility, efficiency, low solution and sample consumption, and low-cost analysis tolerance of complex matrices, which can be processed without extensive pretreatment in comparison to some AAS techniques. The literature presents some strategies to provide good and quicker separation, such as by adding a complexing agent and/or using a different mode of injection (short-end injection procedure, SEIP) [8,9,10]. In SEIP, the separation length is decreased by injecting the sample at the capillary end closest to the detector window (outlet).

Compared to organic compounds, only a few CE applications are reported for the analysis of inorganic metals. Even so, CE is currently considered to be a powerful instrumental technique for inorganic ion separations as a result of extensive research in this field, as can been seen in many reviews [11,12,13,14]. However, ICP-OES or ICP-MS remain the preferred detection techniques [15]. Based on previous work, it is important to use complexing agents to improve separation and increase selectivity and sensitivity in the analysis of inorganic metals. Some inorganic ligands, such as chloride and cyanide, are less applicable as they require tighter control of complexation conditions. Furthermore, the main parameters that affect the separation of metal ions using CZE involve the nature of the complexing reagent, free ligand concentration, and electrolyte pH.

In addition, associated to that described above, the design of experiment (DOE) is an important tool to control the variables of CE, in which the experimental conditions could be varied from one to another to achieve the expected separation [16]. In this context, the aim of this work was to develop a simple and ultra-fast CE method with SIEP mode injection combined with DOE for simultaneous Cd^2+^, Pb^2+^, and Hg^2+^ determination in *Passiflora incarnate* tea. The validated method was applied in *Passiflora incarnate* tea from a local market.

## 2. Materials and Methods

### 2.1. Reagents and Standards

Standard stock solution of lead (Pb) 10,000 mg L^−1^, cadmium (Cd) 10,000 mg L^−1^, and mercury (Hg) 1000 mg L^−1^ were obtained from SCP Science (Montreal, QC, Canada). Standard solutions containing different concentrations of Cd^2+^, Pb^2+^, and Hg^2+^ were prepared by mixing the appropriate amounts of the above concentrates. The solvents used were ethanol and methanol purchased from Dinâmica Química (Indaiatuba, SP, Brazil). Deionized water with resistivity of 18.2 Ω cm from Milli-Q system (Millipore Bedford, MA, USA) was used to prepare all solutions. Imidazole (99%, *w*/*w*) obtained from Merck Millipore (Darmstadt, Germany), 2-hydroxyisobutyric acid (HIBA), and 18-crown-6 (99%, *w*/*w*, Sigma Aldrich^®^, Steinheim, Germany), all with analytical grade purity, were used for background preparation.

### 2.2. Apparatus

All CE experiments were carried out using a capillary electrophoresis system with a diode array detector from Beckman Coulter Instruments, model P/ACETM MDQ (Fullerton, CA, USA), operating at 214 nm. The 32 Karat™ software, also from Beckman, was used to control the instrument and for data acquisition. Samples and standards were hydrodynamically injected (5 s at 0.5 psi) at the short end of a fused silica capillary with polyimide coating (50 cm total length, 10.2 cm effective length, 50 µm i.d.). Before the first use, the capillary was rinsed with aqueous 1.0 mol L^−1^ NaOH for 20 min, followed by ultrapure water for 20 min. Every day before starting the analysis, the capillary was rinsed with running buffer for 5 min.

The separation was accomplished using BGE containing 5 mmol L^−1^ imidazole, 6 mmol L^−1^ HIBA, and 23.3 mmol L^−1^ 18-crown-6 in 3.7 % (*v*/*v*) methanol. A constant voltage of +11.4 kV was applied on the injection side (short end) with temperature fixed at 20 °C.

### 2.3. Standard Solutions for CZE Analyses and Sample Preparation

For the method development and quantification, stock solutions of 4 µg mL^−1^ in ultrapure water were prepared for each analyte. The working standard solutions with suitable concentration ranges were performed for Cd^2+^, Pb^2+^, and Hg^2+^ by the appropriate dilutions of the stock solutions with ultrapure water, which were stored in a refrigerator (+4 °C) up to analysis.

The leaves of *Passiflora incarnata* were purchased from pharmacies in the metropolitan region of São Paulo. The tea was prepared as follows: a volume of 50 mL of Milli-Q water (ultrapure water) was heated to a temperature between 40 and 50 °C, to which approximately 1 g of tea was added and kept for 10 min at this temperature. Following this, the tea was allowed to cool down, then it was filtered with qualitative filter paper and analyzed by the CE method without sample preparation.

### 2.4. Experimental Design

DOE were performed using Minitab^®^ software (State College, PA, USA). The screening stage aims to achieve rapid and acceptable separations for all analytes under investigation. The DOE was based on the full factorial design of the crucial factors at two levels.

### 2.5. Analytical Performance

Linearity, precision, accuracy, limit of detection (LOD), and limit of quantification (LOQ) have been considered in the validation. First, tea samples without the presence of analytes and spiked samples were analyzed by the CE method to verify the selectivity of the method: that is, if possible, the interferences would not affect the analyses, with the same migration time for all analytes. In this study, the linearity of the method was determined by the analytical curve, with six concentrations ranging from 0.2 to 4.0 µg mL^−1^ for Cd and 1.5 to 4.0 µg mL^−1^ for Pb. Based on the linear regression method, the correlation coefficient (r) was also calculated, making it possible to assess the quality of the obtained curve. To confirm the linearity of the method, the study of the lack of fit was also carried out. Repeatability (relative standard deviation, %RSD) and accuracy (relative error, RE%) were assessed using three concentrations (low, middle, and high concentrations) and six replicates from solutions at concentrations of 0.3, 0.5, and 1.0 µg mL^−1^. LOD was established experimentally (n = 5) by the lowest concentration of the analyte detected, but not quantified by the analytical method, while LOQ was determined experimentally by analyzing known concentrations of the analytes and evaluating the smallest amount that could be quantified from each one with acceptable precision and accuracy (≤20%) under experimental conditions (n = 6).

## 3. Results

### 3.1. CE Method Development

The matrix DOE was constructed to optimize not only the BGE composition but also some instrumental variables, such as temperature and voltage, as described in Table 1. On the other hand, with SEIP, the shortest migration time (2.4 min) and a better peak shape were observed, except for Hg^2+^, as shown in Figure 1. Thus, the same full factorial design shown in Table 1 was employed with the short-end injection procedure, omitting the Hg^2+^ peak. As a result, methanol had affected the peaks’ shape, the separation between analytes and interference, and time analysis, which was shorter than that without methanol concentration (Figure 2). This result is in accordance to Pareto analysis and the main effects and desirability, which show that methanol is the variable that most affects the separation, as shown in Figure 3A and B, respectively. Therefore, from this full factorial design, 18-crown-6 and voltage (kV) were fixed (23.3 mmol L^−1^ and 11.4 kV) and a fine-tuning was performed through employing the response surface methodology (Table 2) evaluating only methanol concentration and temperature. Figure 3C presents the optimum values for these two factors for separation of Cd^2+^ and Pb^2+^ from their adjacent interfering. Figure 4A and B show the good separation of Cd^2+^ and Pb^2+^ in ultrapure water and *Passiflora incarnata* tea in the final optimized condition.

### 3.2. CE Method Performance

The selectivity led to a verification that the interferents not eluted in the same migration times of the analytes, guaranteeing the clear identification of the electropherograms signals for all metals. Calibration curves (n ≥ 6) of heavy metals were performed in triplicate in the range from 0.2 to 4.0 µg mL^−1^ for Cd^2+^ and 1.5 to 4.0 µg mL^−1^ for Pb^2+^. The method proved to be linearly ranged, with values of linear correlation coefficient of 0.9986 and 0.9954 for Cd^2+^ and Pb^2+^, respectively. Linearity was confirmed by the ANOVA test of lack of fit, which provided results that proved that the instrumental response varies proportionally with the concentration of analytes in the matrix (*F_tab_* > *F_cal_*, *p*-value > 0.05 and RSD% of the slope of the calibration curve below 15%). The calibration parameters of the proposed method are summarized in Table 3. The concentration of 0.2 µg mL^−1^ for Cd^2+^ and 1.5 µg mL^−1^ for Pb^2+^ were considered the LOQ for both analytes, since this was the lowest concentration at which they could be quantified with precision and accuracy below 20%. The values referring to repeatability and accuracy are also shown in Table 3. RSD% and RE% values below 15% were obtained for all concentrations evaluated, indicating that the analytical method presented adequate precision and accuracy.

## 4. Discussion

Many variables are important when obtaining CE separations, such as temperature, voltage, buffer pH, BGE composition, and buffer modifiers. Special attention needs to be paid to inorganic cations’ determination/detection by CE [17,18,19,20]. Most inorganic cations do not show absorption in the UV-Vis region, thus a chromophore addition in BGE is needed, such as imidazole [21]. In addition, mobility values in this kind of ions can result in similar migration times that lead to a poor resolution between them. Thus, a better inorganic cations separation can be achieved by changing their mobilities when complexing agents are used in a BGE. The literature recommends the use of 18-crown-6, individually or combined with another complexing agent, as well as with HIBA and with some organic modifiers as methanol [21]. Therefore, in this study, 18-crown-6 was evaluated, combined or not with methanol, as a BGE modifier, after some initial experiments with lactic acid, which did not promote the separation of Cd^2+^, Pb^2+^, and Hg^2+^ (Appendix A).

The first optimization was conducted by CE normal mode with a long-end injection (effective length of 38.8 cm, 50 cm length total, 50 µm i.d.) with a full factorial with a central point. BGE composition and some instrumental variables, such as temperature and voltage, were evaluated. As a result, longer time analysis (>8 min) and bad peak shapes were obtained (Appendix A). These results were due to the electrostatic interaction between Cd^2+^, Pb^2+^, and Hg^2+^, and silanol groups (pka around 3.6), which are negatively charged in BGE pH (around pH 6). By using the SEIP, a shorter migration time (2.4 min) and better peak shape were achieved, except for in Hg^2+^.

Thus, the conditions of the optimized and validated method were as following: indirect detection at 214 nm, hydrodynamic short-end injection (0.5 psi for 5 s), effective length of 10.2 cm (total capillary length: 50 cm), voltage at +11.4 kV, and BGE containing 5 mmol L^−1^ imidazole, 6 mmol L^−1^ HIBA, and 23.3 mmol L^−1^ 18-crown-6, containing 3.8% methanol at temperature of 20 °C. Using these conditions, all separation parameters showed acceptable values, with Rs > 1.5 and N > 3000. The method showed be selective, linear, precise, and accurate, and able to analyze real samples of *Passiflora incarnata* tea. It is important to mention that the method was capable to analyze samples of *Passiflora incarnata* tea without sample preparation. The *United States Pharmacopoeia* chapter <2232> Elemental Contaminants in Dietary Supplements establishes the maximum limits for Cd^2+^ and Pb^2+^ of 0.5 and 1.0 µg g^−1^, respectively [6], therefore making it possible for this method to be directly applied for Cd^2+^ determination and Pb^2+^ detection in *Passiflora incarnata* tea samples.

## 5. Conclusions

Short-end injection capillary electrophoresis has been shown to be a powerful separation method for the determination of heavy metals. This strategy combined with BGE composition alteration associated to a DOE was a fast method, since it did not require sample preparation and the analysis time was less than 3 min; it is ecologically attractive, because negligible volumes of methanol have been used, and simple and inexpensive because it showed low cost and a versatile technique compared to other instrumental techniques. The validated method showed good selectivity, sensitivity, and linearity, and an LOQ for Cd^2+^ in agreement to USP. In this matrix, the method has not been presented as reaching a suitable LOQ for Pb^2+^ according to the US Pharmacopeial Convention’s limit requirements for elemental impurities. However, it has been selective for Pb^2+^, and this elemental impurity can be detected successfully. The method has been applied in tea samples for the determination of the elemental impurities Cd^2+^ and Pb^2+^ through direct injection of *Passiflora incarnate* tea in CE-DAD. Finally, it is important to mention that a suitable sample preparation method can improve sensibility and selectivity, which can be also extended to other matrices.

## Figures and Tables

**Figure 1 ijerph-19-15994-f001:**
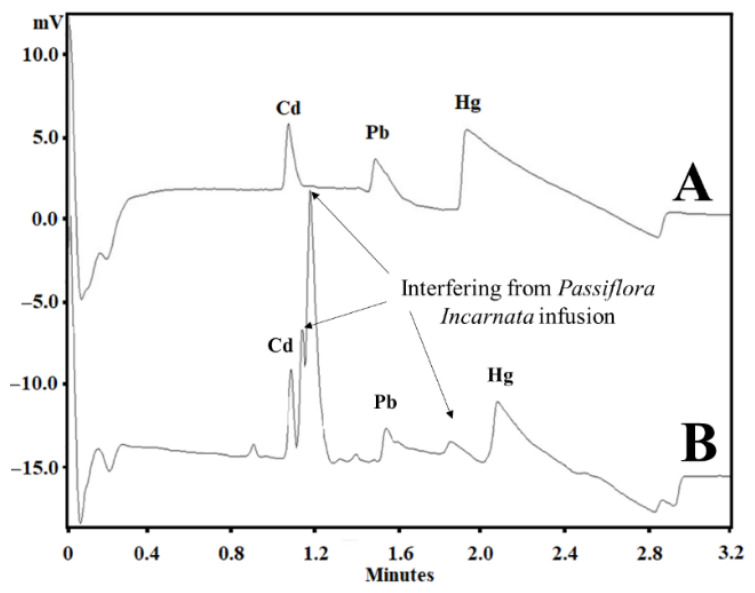
(**A**) Electropherogram of Cd^2+^, Pb^2+^, and Hg^2+^ standards at 6, 3, and 6 µg mL^−1^, respectively, in ultrapure water; (**B**) Electropherogram of *Passiflora incarnata* tea spiked with Cd^2+^, Pb^2+^, and Hg^2+^ standards at 6, 3, and 6 µg mL^−1^, respectively. Other conditions: +30 kV, 20 °C, BGE with 5 mmol L^−1^ imidazole, 6 mmol L^−1^ HIBA, 20 mmol L^−1^ 18-crown-6, 12 cm effective length, 50 cm total length capillary, hydrodynamic short-end injection (0.5 psi by 5 s), and indirect detection at 214 nm.

**Figure 2 ijerph-19-15994-f002:**
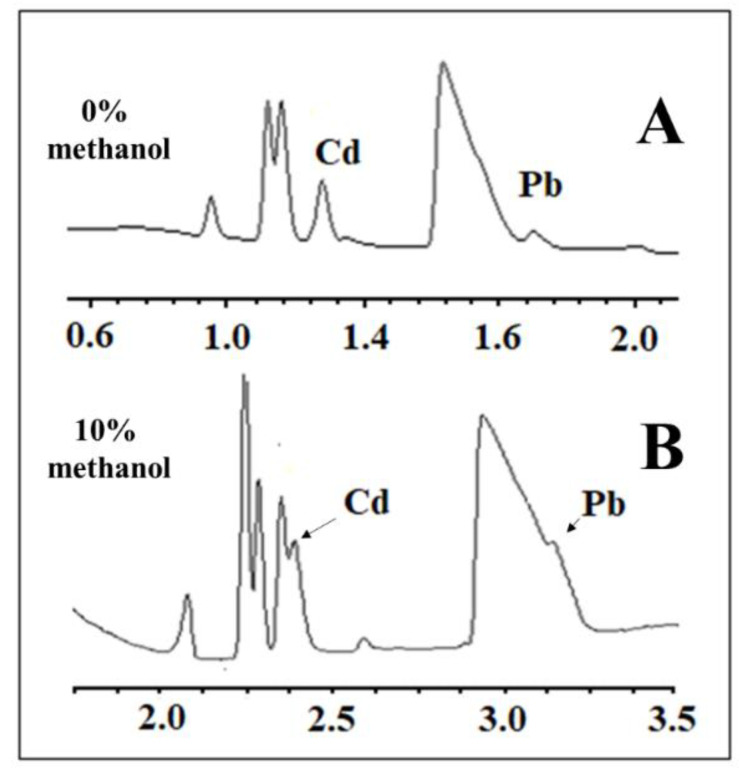
Electropherogram of Cd^2+^, Pb^2+^, and Hg^2+^ standards at 2 and 4 µg mL^−1^, respectively, in *Passiflora incarnata* tea samples using the conditions of (**A**) Experiment 7 (0% methanol and temperature at 15 °C) and of (**B**) Experiment 6 (10% methanol and temperature at 22.5 °C) from Table 1. Other conditions: indirect mode, 12 cm effective length, 50 cm full-length capillary, BGE with 5 mmol L^−1^ imidazole, 6 mmol L^−1^ HIBA, hydrodynamic short-end injection (0.5 psi by 5 s), and detection at 214 nm.

**Figure 3 ijerph-19-15994-f003:**
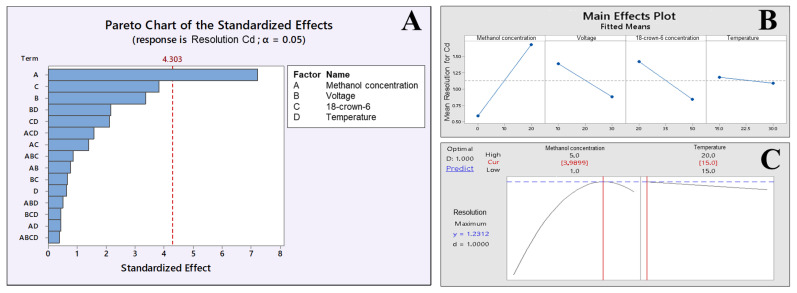
(**A**) Pareto, (**B**) main effects, and (**C**) desirability (methanol concentration and temperature). Graph obtained through DOE by Minitab^®^ software to optimize the separation by CE for Cd^2+^ and Pb^2+^ determination in *Passifora incarnata* tea.

**Figure 4 ijerph-19-15994-f004:**
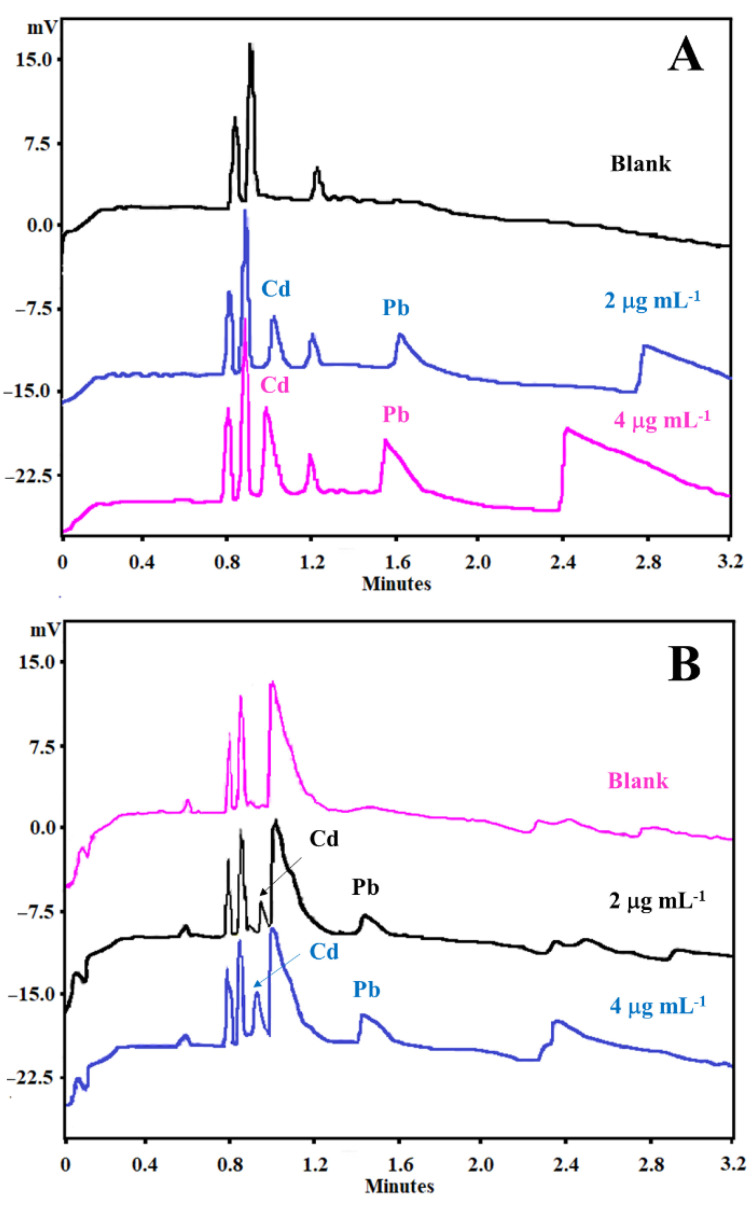
Electropherograms of Cd^2+^ and Pb^2+^ standards at 2 and 4 µg mL^−1^ spiked in (**A**) ultrapure water (black line, blank; blue line, 2 µg mL^−1^; pink line, 4 µg mL^−1^) and (**B**) *Passiflora incarnata* tea samples (pink line, blank; black line, 2 µg mL^−1^; blue line, 4 µg mL^−1^). Conditions: indirect mode, 12.2 cm effective length, 50 cm full-length capillary, BGE with 5 mmol L^−1^ imidazole, 6 mmol L^−1^ HIBA, 32.5 mmol L^−1^ 18-crown-6, 3.8% methanol, temperature at 20 °C, hydrodynamic short-end injection (0.5 psi by 5 s), and detection at 214 nm.

**Table 1 ijerph-19-15994-t001:** Levels used for the implementation of DOE using four factors.

Factor		Level		
		−	+	0*
1	Temperature/°C	15	30	22.5
2	Voltage/kV	10	30	20
3	18-crown-6 concentration/mmol L^−1^	20	50	35
4	methanol percentage/% (*v*/*v*)	0	20	10

* Central point was analyzed three times. − Lower level; + Higher level; 0 Central point.

**Table 2 ijerph-19-15994-t002:** Levels used for the implementation of DOE using two factors.

Factor		Level		
		−	+	0*
1	Temperature/°C	15	20	17.5
2	methanol percentage/% (*v*/*v*)	1	5	3

* Central point was analyzed only once. − Lower level; + Higher level; 0 Central point.

**Table 3 ijerph-19-15994-t003:** Linearity, ANOVA lack of fit, limit of quantification, precision, and accuracy of Cd^+2^ and Pb^+2^ in *Passiflora incarnata* tea by the developed method.

Analytes	Cd^+2^			Pb^+2^		
Linearity						
Linear Equation ^a^	*y* = 3209.4*x* − 469.61			*y* = 736.34*x* − 714.25		
Correlation coefficient (r)	0.9986			0.9954		
Range/µg mL^−1^	0.2–4.0			1.5–4.0		
RSD% ^b^	5.32			6.77		
**ANOVA Lack of fit**						
*F*-value ^c^	1.35			1.63		
*p*-value ^d^	0.22			0.15		
**Limit of detection**						
	0.06			0.45		
**Limit of quantification**						
Nominal concentration/µg mL^−1^	0.20			1.50		
Analyzed concentration/µg mL^−1^	0.18			1.45		
RSD% ^e^	2.57			5.27		
RE% ^f^	−10.0			−3.33		
**Precision/Accuracy**						
**Repeatability/% (n ^g^ = 3)**						
Nominal concentration/µg mL^−1^	0.30	0.50	1.00	1.50	2.50	3.50
Analyzed concentration/µg mL^−1^	0.34	0.45	1.11	1.43	2.68	3.78
Precision/RSD% ^h^	4.97	14.49	12.34	2.46	6.47	7.59
Accuracy/RE% ^i^	13.33	−10.00	11.00	−4.67	7.20	8.00

^a^ Calibration curves were determined in triplicate (n = 3); *y* = A*x* + B, where *y* is the peak area of analyte, A is the slope, B is the intercept, and x is the concentration of the measured solution in ppm; ^b^ RSD%, relative standard deviation of the slope of the calibration curve; ^c^
*F_crit_* < *F_cal_* = 3.467; ^d^
*p*-value > 0.05; ^e^ RSD, relative standard deviation of the limit of quantification; ^f^ RE% = relative error with mean of three replicates of limit of quantification; ^g^ n = number of determinations for precision and accuracy; ^h^ RSD%, relative standard deviation in percentage for precision; ^i^ RE%, relative error for accuracy.

## Data Availability

Not applicable.

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
