# Peer review of "Short-End Injection Capillary Electrophoresis and Multivariate Analysis for Simultaneous Determination of Heavy Metals in Passiflora incarnata Tea"

_ijerph, 2022, doi:10.3390/ijerph192315994_

Round 1

Reviewer 1 Report

Cations have been analyzed by capillary electrophoresis (CE) since its introduction to the analytical practice and Google Scholar lists about two hundred papers on this topic. Therefore, the approach used in the paper by Klassen et al is not very novel. Still, it has value as an example of a practical application of CE to solving problems in medical plant characterization.

General comment;

There are other methods of cation analysis like atom spectroscopic methods which have about three orders of magnitude lower LOD than CE usually has. In addition, the instrumentation sophistication for both AAS and CE (commercial instrumentation) are at the same level and the former is more robust and less demanding to operators' qualifications. The authors should discuss more thoroughly what are the advantages of CA over AAS (e.g. opportunity to analyze several cations simultaneously when the concentrations in the sample are high enough to meet the LOD-s of CE).

Specific comments:

According to the Abstract for Cd, its  LOD<0,3 µg/g. In Table 3 LOD-s are given in ng/mL. What is the correct value?

It is not clear from the paper how LOD was determined.

Author Response

Answers to Reviewer 1

Cations have been analyzed by capillary electrophoresis (CE) since its introduction to the analytical practice and Google Scholar lists about two hundred papers on this topic. Therefore, the approach used in the paper by Klassen et al is not very novel. Still, it has value as an example of a practical application of CE to solving problems in medical plant characterization.

Thanks for your comments. This work brings some novelty in the analysis of metals by CE in teas that are widely used.

General comment

There are other methods of cation analysis like atom spectroscopic methods which have about three orders of magnitude lower LOD than CE usually has. In addition, the instrumentation sophistication for both AAS and CE (commercial instrumentation) are at the same level and the former is more robust and less demanding to operators' qualifications. The authors should discuss more thoroughly what are the advantages of CA over AAS (e.g. opportunity to analyze several cations simultaneously when the concentrations in the sample are high enough to meet the LOD-s of CE).

Thanks for your recommendation. A better discussion of importance of CE methods has been introduced in the manuscript. However, ICP-optical emission spectroscopy or ICPMS remain the preferred detection techniques, but CE methods has gained a lot of attention in the last few years. This information has been inserted in the manuscript.

Specific comments:

According to the Abstract for Cd, its  LOD<0,3 µg/g. In Table 3 LOD-s are given in ng/mL. What is the correct value?

Thanks for your observation. The abstract and tables have been corrected.

It is not clear from the paper how LOD was determined.

Thanks for your question. This information was improved in the manuscript.

Reviewer 2 Report

A short-end injection capillary electrophoresis method was used to quantify heavy metals (Cd2+ and Pb2+) in Passiflora Incarnata tea samples. The analysis time is less than 3 min. The effective separation of different metal ions was achieved by the optimization of experimental conditions, including temperature, voltage, methanol percentage and 18-crown-6 concentration. The linear ranges of ion detection were from 0.2 to 4.0 μg mL-1 for Cd2+ and 1.5 to 4.0 μg mL-1 for Pb2+. The MS can be accepted for publication after the following revision.

The introduction lacks a summary of the work on the use of CE methods for the detection of metal ions and needs to be cited and discussed in detail.  Has the short-end injection procedure been used for the CE detection of metal ions in similar works? the short-and injection procedure needs to be briefly described. What is the separation efficiency?

The description about quantitative analysis is quite short.   In addition, the information about the extract experiments is not provided sufficiently.  Whether the extraction conditions have an effect on the test results? The use of fused uncoated capillaries requires consideration of the effect of metal ion adsorption on quantitative analysis results, although the analysis time is very short. The results of quantitative analysis need to be verified by other methods, such as HPLC. 

Author Response

Answer to reviewer 2

A short-end injection capillary electrophoresis method was used to quantify heavy metals (Cd2+ and Pb2+) in Passiflora Incarnata tea samples. The analysis time is less than 3 min. The effective separation of different metal ions was achieved by the optimization of experimental conditions, including temperature, voltage, methanol percentage and 18-crown-6 concentration. The linear ranges of ion detection were from 0.2 to 4.0 μg mL-1 for Cd2+ and 1.5 to 4.0 μg mL-1 for Pb2+. The MS can be accepted for publication after the following revision.

Thanks for your comments. We have made changes to improve the quality of the manuscript.

The introduction lacks a summary of the work on the use of CE methods for the detection of metal ions and needs to be cited and discussed in detail.

Thanks for your comments. The introduction has been improved.

Has the short-end injection procedure been used for the CE detection of metal ions in similar works? the short-and injection procedure needs to be briefly described.

Thanks for your question. In SEIP, the separation length is decreased by injecting the sample at the capillary end closest to the detector window (outlet). This is a strategy widely used in capillary electrophoresis. This information has been inserted in the manuscript.

What is the separation efficiency?

Thanks for your question. This information has been introduced in the manuscript.

The description about quantitative analysis is quite short.

Thanks for your observation. The description of quantitative analysis has been improved.

In addition, the information about the extract experiments is not provided sufficiently.

Thanks for your observation. The leaves of Passiflora incarnata were purchased from pharmacies of the metropolitan region of São Paulo. The tea was prepared as follows: a volume of 50 mL of Milli-Q water (ultrapure water) was heated to a temperature between 40 and 50 ºC, to which approximately 1 g of tea was added and kept for 10 min. at this temperature. After that, there was a wait until the tea cooled down, and then it was filtered with qualitative filter paper, and then analyzed by CE method. This information has been inserted in the manuscript.

Whether the extraction conditions have an effect on the test results?

Thanks for your question. The tea has been obtained and then analyzed without sample preparation, passthrough only by simple filtration. This information has been introduced in the manuscript.

The use of fused uncoated capillaries requires consideration of the effect of metal ion adsorption on quantitative analysis results, although the analysis time is very short.

Thanks for your observation. The correct information is a fused silica capillary with polyimide coating. This information has been corrected.

The results of quantitative analysis need to be verified by other methods, such as HPLC.

Thanks for your comments. The method has been properly validated for this application and its verification it is not necessary and does not improve consistently the work.